# COVID-19, Long COVID Syndrome, and Mental Health Sequelae in a Mexican Population

**DOI:** 10.3390/ijerph19126970

**Published:** 2022-06-07

**Authors:** Jesús Maximiliano Granados Villalpando, Humberto Azuara Forcelledo, Jorge Luis Ble Castillo, Alejandro Jiménez Sastré, Isela Esther Juárez Rojop, Viridiana Olvera Hernández, Fernando Enrique Mayans Canabal, Crystell Guadalupe Guzmán Priego

**Affiliations:** 1Cardiometabolism Laboratory, Research Center, División Académica de Ciencias de la Salud (DACS), Universidad Juárez Autónoma de Tabasco (UJAT), Villahermosa 86040, Mexico; mgranvilla@outlook.com; 2Instituto de Seguridad Social del Estado de Tabasco (ISSET), Villahermosa 86000, Mexico; ajimenezsastre@hotmail.com (H.A.F.); crystell.guzman@ujat.mx (F.E.M.C.); 3Metabolic Disease Biochemistry, Research Center, División Académica de Ciencias de la Salud (DACS), Universidad Juárez Autónoma de Tabasco (UJAT), Villahermosa 86040, Mexico; jblecastillo@hotmail.com (J.L.B.C.); viridiana.olvera@ujat.mx (V.O.H.); 4Center of Medical Specialitites, Instituto de Seguridad Social del Estado de Tabasco (ISSET), Villahermosa 86000, Mexico; alejandro.jimenez@ujat.mx; 5Lipid Metabolism, Research Center, División Académica de Ciencias de la Salud (DACS), Universidad Juárez Autónoma de Tabasco (UJAT), Villahermosa 86040, Mexico; isela.juarez@ujat.mx

**Keywords:** COVID-19, mental health, Long COVID, anxiety, depression, stress, sequelae

## Abstract

The COVID-19 pandemic is currently a worldwide threat and concern, not only because of COVID-19 itself but its sequelae. The aim of this study was to evaluate whether a relation between COVID-19, Long COVID, and the prevalence of mental health disorders exist. A total of 203 people from Tabasco were included in this study, answering a survey integrated by three dominions: General and epidemiological data, the DASS-21 test (to determine the presence of signs or symptoms suggestive of depression, anxiety, and/or stress) and an exploratory questionnaire about Long COVID syndrome. A descriptive and inferential statistical analysis was made via Microsoft Excel and Graphpad Prism software, evaluating differences through the Mann–Whitney U test and considering *p* < 0.05 as statistically significant. Of the 203 people surveyed, 96 (47.29%) had had COVID-19 and 107 (52.71%) had not; from the ones that had COVID-19, 29 (30.21%) presented mental health disorders and 88 (91.66%) presented at least one symptom or sign of Long COVID syndrome; meanwhile, 31 (32.29%) presented 10 or more symptoms or signs. From the comparison between the population with previous mental health disorders and COVID-19 and those without background disorders or COVID-19, the results were the following: 27.58% vs. 16.82% presented severe depression, 48.27% vs. 17.75% presented severe anxiety, and 27.58% vs. 20.56% presented severe stress. A high prevalence of mental health effects was observed in patients without COVID-19 and increased in the population with Long COVID syndrome and previous mental health disorders.

## 1. Introduction

Identified in December 2019 in the municipality of Wuhan, China, the Novel-Coronavirus (SARS-CoV-2) has since become the main concern of health systems worldwide [1,2]. Aacknowledged as a public health emergency of international concern on January 31 of 2020 and having reached 16.4 million cases, it achieved the status of a pandemic on 11 March 2020 [3,4,5].

Moreso, the COVID-19 pandemic added itself to a large array of transmissible and non-transmissible diseases that were already considered a worldwide danger and concern, generating a syndemic [6,7]. However, not only has physical health been affected but mental health as well, as seen in other various pandemics and epidemics such as the 2003 SARS epidemic, 2011 Influenza pandemic, and 2015 MERS-CoV epidemic [8,9,10].

For instance, some research [11,12,13] shows evidence of anxiety, post-traumatic stress disorder, depression, and fatigue as the main psychiatric morbidities among survivors in the post-illness stage of previous coronavirus epidemics.

On the other hand, it has been demonstrated that up to 87.4% of COVID-19 recovered patients have had several sequelae such as fatigue, dyspnea, stress, depression, anxiety, mental fog, pain, sleep deprivation, cognitive difficulties, and general confusion [14,15,16,17,18,19].

This cluster of persistent symptoms and signs normally does not require internment and is known as Long COVID or Post-COVID Syndrome [20,21]. This syndrome has shown that is a very diverse and unpredictable disease that affects a major amount of COVID-19 recovered patients, presenting a particular challenge in medicine and public health [22,23,24].

In this sense, the COVID-19 pandemic along with confinement, social problems, work, uncertainty about the future, and social status have become important stressors, increasing the so-called “post-pandemic” wave of mental health consequences of COVID-19 [25].

Consistently and invariably, COVID-19 and Long COVID syndrome, as said by other researchers, have increased the incidence of mental health disorders, such as depression, stress, anxiety, and PTSD, having a great impact not only on physical health aspects but on the economic and psychosocial as well [26,27,28,29].

The motivation for this research to be conducted comes from the acknowledgment of psychiatric disorders and mental health conditions as an increasing worldwide public health problem. So much so that the WHO launched the “WHO Special Initiative for Mental Health (2019–2023): Universal Health Coverage for Mental Health” in 2019 in order to increase awareness [30]. However, the COVID-19 pandemic along with quarantine and isolation measures has certainly contributed to the increase in these mental health conditions [31].

The aim of this study was to evaluate whether a relation between COVID-19, Long COVID, and the prevalence of mental health disorders exist.

## 2. Materials and Methods

### 2.1. Study Design and Data Sources

The present observational analytical cross-sectional study was part of a larger project addressing the mental health of COVID-19 patients between the Juarez Autonomous University of Tabasco (UJAT) and a general Hospital in Tabasco, called the “Psychological effects evaluation in patients diagnosed with COVID-19 in a General Hospital in Tabasco” and funded by the Science and Technology Council of Tabasco State (CCYTET). Stress, anxiety, and depression in their various degrees of severity as well as the presence of Long COVID syndrome (post-COVID syndrome or Hauling COVID) were investigated.

### 2.2. Study Setting

This study was conducted in June 2021, during the third COVID-19 wave, via an online questionnaire distributed to various inhabitants of Tabasco, Mexico. Recruitment started on 14 June 2021 and ended on 26 June 2021.

### 2.3. Participants and Procedures

Participants in the study were a random non-probability and purposive sampling approach for convenience consisting of volunteer targeted sampling. In order to be eligible for the study, all surveyed had to accept informed consent. Both genders were admitted, their age had to be between 18 and 64 years old, only inhabitants of Tabasco, Mexico were accepted, and the participants had to answer all sections of the questionnaire.

In order to conduct the analyses, two groups were defined: a healthy control group with no previous SARS-CoV-2 infection and a second group consisting of those with previous SARS-CoV-2 infection which was divided between those with previous mental health disorders and those without. Those surveyed were not compensated for participation in this study.

### 2.4. Measures, Variables, and Data Collection

All participants answered a general data survey for typification and epidemiological classification including sex, age, previous COVID-19 diagnosis, and previous mental health disorders.

In order to assess mental health, the 21-item Depression, Anxiety, and Depression Scales (DASS-21 test) was utilized, consisting of a tripartite model of emotion, measured with a Likert-like scale (in which 0 was never, 1 sometimes, 2 many times, and 3 almost always); low positive affectivity, psychophysiological agitation, and negative affectivity possessed excellent internal consistency, having reliability with Cronbach’s alpha values of 0.81, 0.89, and 0.78 for the subscales of depression, anxiety, and stress, respectively [32].

All participants previously diagnosed with COVID-19 were given an extra exploratory Long COVID syndrome questionnaire, in order to assess the prevalence of Long COVID signs and symptoms, consisting of 27 items in which the most common were grouped into five different clusters: constitutional (exhaustion, fatigue secondary to excessive effort, arthralgia, myalgia, constipation, diarrhea, and cephalea), respiratory (dyspnea, anosmia, drowning sensation, persistent cough, and chest pain), blood vessel associated (petechiae, ecchymosis, high blood pressure, palpitations, and night sweats), endocrinological (alopecia, polydipsia, polyphagia, polyuria, high blood sugar, and acanthosis nigricans), and mental health (mental fog, forgetfulness, inconsistent sleep pattern, and sleep deprivation) assessed with dichotomic variables (yes or no).

The variables hereby acknowledged were mental health symptoms and/or signs, the severity of such symptoms and/or signs determined by their scores, Long COVID syndrome, and mental health antecedents.

Data collection was made via the survey previously mentioned, all data was gathered on Microsoft Excel (2021 version) designing various specific clusters of information: COVID-19 patients with mental health antecedents, COVID-19 patients with mental health antecedents and Long COVID, COVID-19 patients with Long COVID but without previous mental health disorders, non-COVID-19 participants, and consequential subgroups with the determined depression, stress, and anxiety scores and their severities.

### 2.5. Statistical Analysis

Data were analyzed using Microsoft Excel (2021 version) and Prism (Version 9; Graphpad). The continuous variables were assessed for normality using the Shapiro–Wilk test and they all showed significant deviation from the normal distribution (*p* < 0001). Due to this, non-parametric inferential tests were used: the Mann–Whitney U test was used to analyze group differences between the variables previously mentioned. A cut-off score of >10, >8, and >15 was used for DASS-21 to determine the presence of depression, anxiety, and stress, respectively. Meanwhile, a cut-off score of >21, >15, and >26 was used to determine the presence of severe depression, severe anxiety, and severe stress, respectively. These cut-off scores are the standardized cut-off points [33,34]. A *p*-value of <0.05 was considered statistically significant. Tables were used to present these findings.

## 3. Results

The sample consisted of 203 surveyed individuals, of which 131 (64.53%) were female and 72 (35.46%) were male; those 203 were divided into three different age groups: 18 to 25 years with 91 (44.82%) surveyed, 26 to 39 years with 75 (36.94%), and 40 to 64 years with 37 (18.22%) surveyed.

Among the 203 surveyed, 96 were COVID-19 recovered patients, of which 66 (68.75% were females and 30 (31.25%) were males; among these, 32 belonged to the first age group (18 to 25 years old), 43 to the second age group (26 to 39 years old), and 21 to the third group (40 to 64 years old). Of the 96 COVID-19 recovered patients, 29 (30.21%) had mental health antecedents while 67 (69.79%) did not.

The presence of signs and symptoms suggestive of Long COVID syndrome was evaluated via an exploratory questionnaire from the 96 recovered COVID-19 patients: 88 (91.66%) presented at least one symptom or sign, 87 (90.62%) presented two or more, 80 (83.33%) presented three or more, 67 (69.79%) presented five or more, and 31 (32.29%) presented ten or more signs and/or symptoms suggestive of Long COVID. In this sense, the most common signs and symptoms were exhaustion (58, 60.4%), cephalea (51, 53.12%), sleep deprivation (48, 50%), and inconsistent sleep pattern (41, 42.7%). (Table 1).

Of the 29 COVID-19 recovered patients that had mental health antecedents, 28 (96.55%) presented at least one sign or symptom suggestive of Long COVID, 27 (93.1%) two or more signs and/or symptoms, 26 (89.65%) three or more, 22 (75.86%) five or more, and 12 (41.37%) presented ten or more signs and/or symptoms suggestive of Long COVID.

The most common signs or symptoms among those 29 COVID-19 recovered patients with mental health antecedents were: sleep deprivation (20, 68.96%), inconsistent sleep pattern (18, 62.06%), exhaustion (18, 62.06%), cephalea (18, 62.06%), memory issues (17, 58.62%), and mental fog (14, 48.27%) (Table 2).

In order to assess mental health, the DASS-21 test was utilized. Of the 107 surveyed that had no COVID-19 antecedents, 53 (49.53%) had data suggestive of at least one mental disorder, 40 (37.38%) had more than one mental health disorder, and 25 (23.36%) had data suggestive of all three mental disorders assessed hereby, resulting in a total of 40 (37.38%) that presented data suggestive of depression, 39 (36.44%) had data suggestive of anxiety, and 42 (39.25%) had data suggestive of stress.

Of the 96 COVID-19 recovered patients, 60 (62.25%) presented at least one mental health disorder, 38 (39.58%) two or more mental health disorders, and 24 (25%) presented all three mental health disorders, resulting in 39 (40.62%) presenting data suggestive of depression, 46 (47.91%) with data indicative anxiety, and 42 (37.5%) suggestive of stress.

Of the 29 COVID-19 patients that had previous mental health disorders, 22 (75.86%) presented suggestive data of at least one mental health disorder, 15 (51.72%) presented data of at least two mental health disorders, and 11 (37.93%) presented all three mental health disorders, resulting in a total of 14 (48.27%) that presented data suggestive of depression, 19 (65.51%) with data suggestive of anxiety, and 15 (51.72%) with data indicative of stress.

In the comparison of different groups assessing the scores of depression, anxiety, and stress among COVID-19 recovered patients and non-COVID-19 surveyed, it was found the differences were strong if not statistically significant (Table 3), demonstrating, therefore, that the presence of COVID-19 and previous mental health disorders increases the chance of having depression, anxiety, and/or stress (Table 4).

## 4. Discussion

The present study has demonstrated that there is a relation between COVID-19, Long COVID syndrome, and mental health disorders, showing that even in supposedly healthy patients with no COVID-19 antecedents the prevalence of mental health disorders is elevated and, very worrying, there exists a statistically significant association between depression, anxiety, and stress scores and COVID-19 along with previous mental health disorders, depression, and anxiety scores and COVID-19 as well as a clear tendency towards significance between stress and COVID-19.

This is consistent with several studies which highlight a stretch relation between mental health and COVID-19 [25,26,27]. Furthermore, the high prevalence of mental health disorders in the general population supports the studies that said that, invariably, the COVID-19 pandemic was going to increase depression, anxiety, and stress in the global population, having a great psychosocial impact [28,29].

This work highlights the high prevalence of Long COVID signs and symptoms, showing that 91.66% of COVID-19 recovered patients had at least one, and almost a third (32.29%) had ten or more. Among COVID-19 recovered patients with previous mental health disorders, the situation was more severe, with 96.55% of patients having at least one sign or symptom of Long COVID, and practically two out of five (41.37%) having ten or more, which is consistent with other studies [14,15,16,17,18,19].

Additionally, from the presence or absence of Long COVID syndrome, a clear pattern of severity in suggestive signs and symptoms of depression, anxiety, and stress and the presence of COVID-19 and mental health antecedents was found, for example, in the contrast between the non-COVID-19 surveyed with severe depression (16.82%), severe anxiety (17.75%), and severe stress (20.56%) and COVID-19 recovered patients with mental health antecedents with severe depression (27.58%), severe anxiety (48.27%), and severe stress (27.58%).

It is evident that there is a clear relationship between Long COVID syndrome and mental health disorders, whether there are mental health antecedents or not, making this population (with Long COVID and mental health antecedents) a risk group to which special attention, tracing, and care must be given.

The main limiting factor in this study was the presence of the COVID-19 pandemic itself which modified the day-to-day activities of the involved personnel and of the surveyed, restricting the method and promptness of the survey, data collection, data interpretation, and data analysis, and deriving some weaknesses in the research such as the inability to further explore the surveyed and to conduct a thoughtful interview to identify behaviors and clinical data that could be useful to future research, such as the COVID-19 severity of those recovered, due to the lack of a critical and objective analysis of the surveyed and their medical history and chemical, clinical, and radiologic markers to correctly classify and interpret. Furthermore, the magnitude of the sample is a highlighted weakness, being so small that it does not reach statistical significance in some variables; however, there is a clear tendency towards significance as has been mentioned before.

Some other limitations of the study include the lack of more variables such as employment status, occupation of the subjects, educational level, marital status, residential region, type of habits, comorbidities, or monthly household income that have also been associated with an increase in psychiatric disorders [33,34].

Moreover, there is a potential confounding variable because of the absence of a non-COVID-19 group with previous mental health disorders, resulting in the possibility of high depression, anxiety, and stress scores being due to only the antecedent of previous mental disorders and not necessarily the combination of both COVID-19 and previous mental disorders.

Additionally, there is an inability to determine whether the mental health affection pertains to Long COVID syndrome or is a consequence of the same, which would require cohort studies focused on this, but, independent of this reason, and as it is mentioned in other studies [32,35,36], this is a key opportunity to establish policies and protocols for psychological and psychiatric attention in health personnel and the general population.

To obtain greater statistical significance and a better data interpretation, it would be necessary to appeal towards prospective cohort studies, have a greater sample and control over more variables and groups, and analyze the evolution of theses in real-time, therefore limiting any bias.

Overall, the results hereby presented are relevant not only to medical or epidemiological ambits but to public health, social studies, and even economic ambits due to the study showing the necessity of integral attention, follow-up, and treatment in every COVID-19-recovered patient, in every patient with mental health antecedents, and the importance of primary and secondary prevention in the general population.

These findings are a call of attention to health systems or focus efforts on vigilance and protection of mental health, recommending: close tracking of COVID-19 recovered patients, with an integral and multidisciplinary focus; mental health promotion with the main focus on primary and secondary prevention of mental health disorders; taking into consideration and attending to all WHO recommendations; and continued researching into the mental health impact of COVID-19 and its relation to Long COVID syndrome.

## 5. Conclusions

COVID-19 is a disease that has afflicted the global population for almost two years in various ambits, public health, mental health, economy, and politics, and is most likely far from disappearing. In addition, the disease itself has caused long term effects on recovered patients, generating a syndrome now known as “Long COVID”, “Hauling COVID”, or “Post-COVID syndrome” which is characterized by the presence of several signs and symptoms that affect physical and mental health.

This syndrome has been related to a high prevalence and incidence of mental health disorders at a global level. In this study, it has been demonstrated that the presence of COVID-19 and Long COVID is related to mental health sequelae. Not only that, but even without COVID-19 or mental health antecedents, the prevalence of mental health disorders in the general population is quite alarming, making mental health a main point of attention, tracing, and prevention in the affected population, health personnel, and general population.

We can conclude, therefore, that the mental health condition of the population during the COVID-19 pandemic is detrimental, especially in subjects who have been infected by SARS-CoV-2 and have had previous mental health disorders.

## Figures and Tables

**Table 1 ijerph-19-06970-t001:** Most common Long COVID syndrome signs and symptoms assessed in COVID-19 recovered population.

	Females		Males		Both Genders	
Mean age	31.9		30.56		31.48	
	*n* (out of 66)	%	*n* (out of 30)	%	*n* (out of 96)	%
Symptoms or Signs						
Constitutional						
Exhaustion	42	63.63	16	53.33	58	60.41
Cephalea	30	45.45	21	70	51	53.12
Fatigue with excessive effort	25	37.87	14	46.67	39	40.62
Myalgia	22	33.33	13	43.33	35	36.45
Arthralgia	22	33.33	10	33.33	32	33.33
Diarrhea	19	28.78	9	30	28	29.16
Constipation	8	12.12	3	10	11	11.45
Blood vessel related						
Night sweats	19	28.78	5	16.67	24	25
Palpitations	17	25.75	4	13.33	21	21.87
High blood pressure	6	9.09	4	13.33	10	10.41
Petechiae	3	4.54	2	6.67	5	5.2
Ecchymosis	3	4.54	1	3.33	4	4.1
Respiratory						
Anosmia	23	34.84	12	40	35	36.45
Persistent cough	12	18.18	6	20	18	18.75
Drowning sensation	16	24.24	8	26.67	24	25
Chest pain	14	21.21	10	33.33	24	25
Dyspnea	19	28.78	10	33.33	29	30.2
Mental						
Mental fog	25	37.87	8	26.67	33	34.37
Memory issues	31	46.96	5	16.67	36	37.5
Inconsistent sleep pattern	32	48.48	9	30	41	42.7
Sleep deprivation	33	50	15	50	48	50
Endocrinological						
Hair loss	25	37.87	6	20	31	32.29
Polydipsia	14	21.21	9	30	23	23.95
Polyuria	10	15.15	8	26.67	18	18.75
Polyphagia	10	15.15	4	13.33	14	14.58
Hyperglycemia	6	9.09	2	6.67	8	8.33
Acanthosis nigricans	2	3.03	0	0	2	2.08

**Table 2 ijerph-19-06970-t002:** Most common Long COVID syndrome signs and symptoms assessed in a COVID-19 recovered population with previous mental health disorders.

	Females		Males		Both Genders	
Mean age	31.61		30.87		31.41	
	*n* (out of 21)	%	*n* (out of 8)	%	*n* (out of 29)	%
Symptoms or Signs						
Constitutional						
Exhaustion	15	71.42	3	37.5	18	62.07
Cephalea	12	57.14	6	75	18	62.07
Fatigue with excessive effort	10	47.61	2	25	12	41.38
Myalgia	10	47.61	2	25	12	41.38
Arthralgia	10	47.61	2	25	12	41.38
Diarrhea	9	42.85	5	62.5	14	48.28
Constipation	2	9.52	1	12.5	3	10.34
Blood vessel related						
Night sweats	7	33.33	1	12.5	8	27.59
Palpitations	8	38.09	0	0	8	27.59
High blood pressure	1	4.76	1	12.5	2	6.89
Petechiae	2	9.52	0	0	2	6.89
Ecchymosis	2	9.52	1	12.5	3	10.34
Respiratory						
Anosmia	8	38.09	2	25	10	34.48
Persistent cough	5	23.8	2	25	7	24.14
Drowning sensation	8	38.09	1	12.5	9	31.03
Chest pain	7	33.33	0	0	7	24.14
Dyspnea	8	38.09	4	50	12	41.38
Mental						
Mental fog	12	57.14	2	25	14	48.28
Memory issues	15	71.42	3	37.5	18	62.07
Inconsistent sleep pattern	9	42.85	1	12.5	10	34.48
Sleep deprivation	15	71.42	5	62.5	20	68.97
Endocrinological						
Hair loss	11	52.38	2	25	13	44.83
Polydipsia	7	33.33	1	12.5	8	27.59
Polyuria	5	23.8	1	12.5	6	20.69
Polyphagia	6	28.57	1	12.5	7	24.14
Hyperglycemia	0	0	1	12.5	1	3.45
Acanthosis nigricans	1	4.76	1	12.5	2	6.89

**Table 3 ijerph-19-06970-t003:** Mental health disorders distribution among COVID-19 recovered patients with and without previous mental health disorders and non-COVID-19 surveyed.

	Non-COVID-19 Surveyed	COVID-19 Recovered without Previous Mental Health Disorders	COVID-19 Recovered with Previous Mental Health Disorders	Mann–Whitney U Test *p*
	6(2,16)	4(2,12)	-	0.57
Depression	6(2,16)	-	10(5,23)	0.024
	-	4(2,12)	10(5,23)	**0.01 ***
	4(0,12)	6(2,12)	-	0.94
Anxiety	4(0,12)	-	14(5,22)	**0.001 ****
	-	6(2,12)	14(5,22)	**0.002 ****
	14(4,22)	10(4,18)	-	0.23
Stress	14(4,22)	-	16(9,28)	0.06
	-	10(4,18)	16(9,28)	**0.006 ***

The median, 25th percentile, and 75th percentile of the depression, anxiety, and stress scores of the different groups are indicated. * *p* < 0.5 *** p* < 0.005. Statistical significance (*p*-value < 0.05) based on the Mann–Whitney U test.

**Table 4 ijerph-19-06970-t004:** Mental health disorders among COVID-19 recovered patients, COVID-19 recovered patients with previous mental health disorders, and non-COVID-19 surveyed.

	COVID-19 Recovered Patients			COVID-19 Recovered Patients with Mental Health Antecedents ^1^			Non-COVID-19 Surveyed		
	Both genders (*n* = 96)	Females (*n* = 66)	Males (*n* = 30)	Both genders (*n* = 29)	Females (*n* = 21)	Males (*n* = 8)	Both genders (*n* = 107)	Females (*n* = 73)	Males (*n* = 34)
Depression	39 (40.62%)	29 (43.93%)	10 (30%)	14 (48.27%)	11 (52.38%)	3 (37.5%)	40 (37.38%)	24 (32.87%)	16 (47.05%)
Severe Depression	14 (14.58%)	10 (15.15%)	4 (13.33%)	8 (27.58%)	6 (28.57%)	2 (25%)	18 (16.82%)	13 (17.8%)	5 (14.7%)
Anxiety	46 (47.91%)	33 (50%)	13 (43.3%)	19 (65.51%)	16 (76.19%)	3 (37.5%)	39 (36.44%)	27 (36.98%)	12 (35.29%)
Severe Anxiety	25 (26.04%)	18 (27.27%)	7 (23.33%)	14 (48.27%)	12 (57.14%)	2 (25%)	19 (17.75%)	14 (19.17%)	5 (14.7%)
Stress	36 (37.5%)	25 (37.87%)	9 (30%)	15 (51.72%)	13 (61.9%)	2 (25%)	42 (39.25%)	29 (39.72%)	13 (38.23%)

The *n* and percentage of prevalence are indicated. ^1^ These surveyed are among the total of COVID-19 recovered patients.

## Data Availability

The datasets used and/or analyzed in the current study are available from the corresponding author on reasonable request.

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
