# Peer review of "COVID-19, Long COVID Syndrome, and Mental Health Sequelae in a Mexican Population"

_ijerph, 2022, doi:10.3390/ijerph19126970_

Round 1

Reviewer 1 Report

Thanks for the Editor for giving me this opportunity to review this article. I am personally very interested in this topic. The research design is appropriate, the methods are adequately described, the results are clearly and briefly presented, and the conclusions are supported by the results.

I have only two suggestions:

1)The introduction should provide more sufficient background.

2)The discussion should be more comprehensive and thorough.

Author Response

"Please dee the attachment"

Reviewer 2 Report

Material and Methods

Why did the authors decide to evaluate 12 days? Is there any suggestion or reference about the days of study? Would it exist any difference if the study was done with more or less days?   Results Lines 135 to 136; "The sample..." Would it be possible to assess the sociodemographic area? It could be that this difference is related to symptomatology. Lines 144 to 153; "The presence..."it is not necessary to explain all values in this paragraph, which was explained on figure 1. Despite the figure 1 is interesting, I recommend to describe the symptomatology in a table, and include the median age and relationship between male and female.   For example:   *LCS Symptoms     Median Age.  Ratio (F:M)   *Long Covid Syndrome   Table 1 It is not necessary to put the p value <0.05, this part is obvious. I recommend to use bold letter in significant p value and delete the supernumeral 1     Table 2; "Mental health..." Would it be possible to describe the relationship of symptoms and gender? For example                       COVID-9 recovered patients                                    %F.          %M.    (F:M) Depression.                 20            20.       1:1      it is necessary to make a test of the homogeneity of the variance and make a normal distribution   Discussion The discussion is interesting and well explained, however I consider is incomplete due to some data that the authors need to evaluate with more variables related to mental health. For example Activity of patients (if this activity is active or passive) Type of work that patients do (office worker, waiter, etc...)   Type of habits (consume of smoke and/or alcohol, or others)   if married/divorced or live alone or with the family, or partner, etc...   Would it be possible to include those variables? Overall This manuscript is interesting and well done, however it is necessary to respond my suggestions and deepen more in the discussion and their variables.

Author Response

"Please see the attachment".

Reviewer 3 Report

The authors report an increased prevalence of mental health effects in patients with long-COVID syndrome and previous mental health disorders. The text needs editing (i.e. without in line 38; space in line 24). Furthermore, the methods need to be more clearly explained why they were included i.e. extra exploratory long COVID syndrome questionnaire. More background information is needed to clarify the motivation for the work presented to the readers and so the conclusion section presents the outcome of the work by interpreting the findings by relating these findings to the motivation stated in the introduction. Furthermore, is there any mention of potential confounding, limitations? 

Author Response

"Please see the attachment".

Round 2

Reviewer 2 Report

The manuscript has improved considerably in comparison to the prior version. Now is more understandable and better explained.